# The Experience of Post-Stroke Pain and The Impact on Quality of Life: An Integrative Review

**DOI:** 10.3390/bs10080128

**Published:** 2020-08-07

**Authors:** Hannah Payton, Andrew Soundy

**Affiliations:** Sport, Exercise and Rehabilitation Sciences, University of Birmingham, Edgbaston, Birmingham B15 2TT, UK; HXP691@student.bham.ac.uk

**Keywords:** review, stroke, pain, quality of life

## Abstract

*Background*: Many people experience post-stroke pain (PSP). It is a long-term consequence of stroke that commonly goes unrecognised and untreated. As a result, an integrative review is needed to identify the primary factors that affect PSP and determine the impact on quality of life (QOL). *Methods*: An integrative review using a quantitatively led data synthesis, supported by qualitative evidence, was conducted. *Results*: Fourteen studies were identified and 2415 (968 females, 1447 males) people were included. Five primary themes were identified as effecting the experience of PSP; anxiety, depression, fatigue, cognitive function and physical function. Anxiety, depression and fatigue increase PSP. Pain, depression, fatigue and reduced physical function lower QOL. *Conclusions*: It is essential that clinicians recognise PSP in order to optimize QOL and function post-stroke. Further research is needed to employ a strategy to identify and objectively quantify PSP and its impact on QOL.

## 1. Introduction

Post-stroke pain (PSP) occurs in around 10–39% of new stroke cases [1,2]. Around 40% of stroke cases experience some form of pain 5 years post-stroke [3]. The most common classification of pain is called central post-stroke pain (CPSP) identified in around 2.7% of total cases [2]. Other classifications include peripheral neuropathy and pain from spasticity or subluxation. Headaches are also a cause of PSP; predominantly tension-type headaches [4]. Around 25% identify that their PSP needs are not met at 5 years [3]. Two-thirds of patients with CPSP report inadequate pain treatment or received no treatment at all [5]. This can occur because patients fail to communicate their pain due to aphasia, neglect and cognitive impairments [6]. This can make it challenging for clinicians to treat. Pain can cause disability secondary to a decrease in physical function, adversely affecting rehabilitation outcomes [7]. Research is needed that is able to consider the factors which influence the experience of PSP. 

Review evidence has identified negative associations between pain and quality of life [8,9,10]. However, there are several weaknesses of the current research including; evidence provided alongside other chronic conditions [8], limited focus on PSP within a broader review topic [9,10] or a lack of focus on qualitative evidence [8,9,10]. Other factors have also been identified as influencing PSP. They include; fatigue, although current evidence for association is mixed [11], mental health although again, evidence has been mixed within the literature. Some evidence identifying a positive association between depression and PSP [6], whilst broader literature has identified a bi-directional relationship between anxiety, depression and PSP [12]. Finally, research has also associated cognition and motor function with PSP [13].

Given the above evidence, it is important that further review-based research is undertaken and brings together understanding on the factors which influence PSP. It is also important that such research documents qualitative evidence and insight to lived experiences [14]. An integrative or mixed studies review may be best placed to achieve this. To the best of the authors’ knowledge, no review has been able to provide an overview across different psychological and physical factors that influence the experience of PSP and identify why such associations occur. The present review was designed to integrate studies that provide further understanding of associations by using different types of evidence.

## 2. Materials and Methods

An integrative review [15] was undertaken in 3 phases: (a) literature search, (b) data appraisal and (c) data synthesis. The literature search was documented using a PRISMA flow diagram [16]. This review is written from the perspective of a subtle realist. This approach recognises common or shared features of reality. These shared features are generated through similar experiences and perceptions. However, it also recognised that each individual’s experience is unique. 

### 2.1. Eligibility Criteria 

An article was included when it satisfied the following eligibility criteria, using the SPIDER tool [17].

*Sample*: Studies has to include individuals who were currently experiencing PSP. There were no exclusions to age because pain is not affected by this variable [1]. There were no exclusions to time post-stroke; due to the complex nature of pain and the varying factors that affect it, there is no direct correlation between these variables [18].

Phenomenon of interest: Articles were selected if they focused on the factors that influence patients’ experience of pain and the impact of these factors on quality of life (QOL). Factors could include physical health, psychological health and level of independence [19]. Pain was considered within different areas of the body and different types of pain were considered including nociceptive, central pain and headache pain. Pain was considered as acute or chronic pain. 

*Design*: Articles were selected with different design types including; (a) quantitative methodologies such as cross sectional study designs, or survey methods, (b) qualitative methodologies included types of grounded theory e.g., social constructionist grounded theory, types of phenomenology e.g., hermeneutic, participatory approaches e.g., community action research, vignette-based studies or ethnography, and (c) any form of mixed methods study. Both single time-point studies and longitudinal studies were included. Only studies written in English were included. Articles were excluded if they were not presented with a methods section that could be critically evaluated (traditional scientific presentation of study findings). Theses were excluded. Case studies were excluded due to the focus on common experiences across people. 

Evaluation: Quantitative studies had to include an objective measure of both pain and QOL or at least three QOL domains, as defined by WHO [19]. Qualitative studies were included where the primary focus of data collection was on patient experience and perception of pain. Mixed method studies were included where the qualitative element had a primary focus on patient experience and perception of pain. Outcome measures included interviews, self-report questionnaires, objective measures of pain and QOL.

*Research type*: Qualitative, quantitative and mixed methods were used.

### 2.2. Literature Search

The electronic search of online databases adhered to guidelines dictated by McGowan et al. [20]. It included, MEDLINE, AMED, CINAHL and PubMED. Databases were searched from inception until January 2020. Key words and Boolean operators were used. The search terms included Experience OR Perception AND Pain OR Chronic pain OR Long-term pain OR Acute pain AND Stroke or Cerebral vascular attack OR CVA AND Quality of life OR Health-related quality of life AND Qualitative OR Quantitative OR Mixed methods. Other electronic searches included www.sciencedirect.com and Google Scholar. Additional articles from reference lists of studies reviewed were identified and included [21]. 

### 2.3. Data Extraction

Demographic details were extracted from studies (see Appendix A).

### 2.4. Critical Appraisal

A quality assessment of qualitative studies was undertaken using the modified consolidated criteria for reporting qualitative research tool (COREQ) [22] adapted from COREQ [23] (see Appendix A). Quantitative studies were assessed using Cochrane’s risk of bias tool [24]. Studies were rated as low, moderately low, moderately high or high risk of bias (see Appendix A).

### 2.5. Data Analysis and Qualitative Synthesis

Common outcome measures were identified, revealing five primary themes (see Appendix A). Quantitative data were summarised using a narrative synthesis of data for each theme. All outcome measures for each theme were identified and findings were assessed and moderated by level of evidence (see Appendix A). Stage 2: Qualitative findings that could support and elaborate on stage 1 findings were identified and integrated after quantitative findings (see Appendix A). Stage 3: A summary was generated of combined findings that had been identified for each theme, with consideration to quality of evidence.

## 3. Results

### 3.1. Search

A total of 86 studies were identified and 14 [1,3,25,26,27,28,29,30,31,32,33,34,35,36] were identified as being appropriate for inclusion. Figure 1 provides a PRISMA diagram. 

### 3.2. Demographics

The total number of participants who were included in the review and had experienced a stroke was 2415 (968/2415, 40% females, 1447/2415, 60% males). The aggregated mean age was 67.3 years (1500/2415, 62% of data represented). There were 414 duplicate participants not included in this number [31,35,36]. The types of stroke included were cerebral infarction (*N* = 1548/2415, 64%), haemorrhagic (*N* = 254/2415, 11%) and unspecified (*N* = 612/2415, 25%). Time post-stroke ranged from one month to five years. Types of pain included nociceptive, CPSP and headaches. Settings included hospital rehabilitation centres (*N* = 4), postal questionnaires (*N* = 4), participants’ homes (*N* = 3) and a telephone interview (*N* = 1). One study used a mix of settings including outpatient clinic, primary care centres, nursing homes, or participants own homes [27]. One study did not specify setting [26]. Studies took place in Sweden (*N* = 6), Norway (*N* = 2), Denmark (*N* = 1), Ireland (*N* = 1), Wales (*N* = 1) Turkey (*N* = 1), Korea (*N* = 1) and Singapore (*N* = 1).

### 3.3. Critical Appraisal 

#### 3.3.1. Between Study Quantitative Appraisal

Quantitative studies identified risks of bias as sampling bias (*N* = 10), attrition bias (*N* = 4), recall bias (*N* = 1), measurement error (*N* = 1), detection bias (*N* = 1), observer bias (*N* = 1) and maturation bias (*N* = 1). Sampling bias was a consistent weakness across studies due to the exclusion of participants with cognitive impairments, speech impairments and communication difficulties, which are common among a stroke population, therefore making samples unrepresentative.

#### 3.3.2. Within Study Quantitative Appraisal

Studies with no risk of bias were identified as low risk (*N* = 0/12). Studies with one risk of bias were identified as moderately low risk (*N* = 6/12). Studies with two risks were identified as moderately high risk (*N* = 5/12), Studies with three or more risks were identified as high risk (*N* = 1/12). The study with the highest risk of bias was Choi-Kwon et al., [25] (see full details in Appendix A).

#### 3.3.3. Between Study Qualitative Appraisal

Two studies were identified as being appropriate for inclusion. Using the COREQ tool, neither study was considered as fatally flawed (weaknesses that compromise quality of data) [37]. Neither study reported on the use of field notes or considered data saturation.

#### 3.3.4. Within Study Qualitative Appraisal

The weakest reporting occurred in domain one (score 1/5) in the study by Widar et al., [36]. Details regarding the experience of the author, established relationships and interviewer characteristics were not commented on. The highest reporting across both studies was domain three (average score 3/3) (see Appendix A).

### 3.4. Synthesis 

Five primary factors were identified as influencing the experience of pain; (a) Depression, (b) Anxiety, (c) Fatigue, (d) Cognitive function, and (e) Physical function. Table 1 provides a summary of this evidence. 

#### 3.4.1. Depression and Pain

##### Scales Used to Assess Depression

Eight studies documented depression. Depression was represented by the Hospital Anxiety and Depression Scale (HADS, *N* = 4), Geriatric Depression Scale (GDS, *N* = 2), Beck Depression Inventory (BDI, *N* = 2), DSM-IV Major Criteria (*N* = 2) and the Risk of Stroke Scale (*N* = 1).

##### Depression and Pain; Quantitative Evidence 

Pain was found to have no association with depression in four studies [26,28,30,32]. In one study where no association was found, a minimal clinically important difference (MCID) was identified. The MCID threshold for the BDI is 17.5% as determined by Button et al. [38]. Results by Sahin-Onat et al. [32] showed a difference of 28% on the BDI between patients with and without pain (based on the median score). The quality of these findings was mixed ranging from moderately low [30,32] to moderately high [26,28].

However, four studies identified a significant association. The first study [27] showed a significant positive association (coefficient B = 2.1) between higher pain intensity and depression (*p* < 0.001). The second study [1], also found a significant association between pain and depression (*p* < 0.001, CI: 3.43 (2.25–5.25)). The third study [33] also found a significant association between pain and depression on both the DSM-IV (*p* = 0.081) and GDS (*p* = 0.009). The fourth study [3] identified higher levels of depression in individuals who reported more frequent pain compared with those reporting less frequent pain using the Risk of Stroke Scale (*p* < 0.001) this finding was supported by the anxiety/depression subscale of the EQ-5D (*p* < 0.001). All significant results were rated as moderately low risk of bias; therefore, it is probable that an association exists.

##### Depression and Pain, Qualitative Evidence 

Qualitative studies supported that pain negatively affects mood and a feeling of depression increases perception of pain [29]. Pain was also described as disheartening and patients found themselves to be angrier when experiencing pain [36].

##### Evidence Summary

Given the above findings, the qualitative evidence adds weight to the significant quantitative findings, suggesting that low (depression) and high (anger) energy unpleasant moods could heighten pain. However, mixed findings mean further research is required. 

#### 3.4.2. The effect of Depression on QOL

##### Scales Used to Assess QOL

QOL was represented by three studies using the Short Form-36 (SF–36, *N* = 2), World Health Organization-BREF Quality of Life Questionnaire (WHOQOL; *N* = 1), EuroQol 5D (EQ5D, *N* = 2) and 15D (*N* = 1).

##### Depression and QOL; Quantitative Evidence

All four studies identified a significant negative association. Kong et al. [28] found that depression had a negative influence on five out of eight domains of the SF-36. Depression was significantly associated with general health (*p* < 0.01), vitality (*p* < 0.01), social functioning (*p* < 0.01), role limitations due to emotional problems (*p* = 0.02), and mental health (*p* < 0.01). Naess et al., [31] found a significant negative association (*r* = −0.50) between depression and Health Related Quality of Life (HRQOL) (*p* < 0.001), as measured by the 15D, but no association on the EQ-5D.

Choi-Kwon et al. [25] found that the presence of depression at three months post-stroke was significantly associated with a low QOL at three years post-stroke (*p* < 0.01). A very small effect size was calculated among patients without depression (Hedges’*g* = (3.49–3.45)/0.47 = 0.09). A small to medium effect size was calculated among patients with depression (Hedges’*g* = (2.89–2.97)/0.21 = 0.38) [39]. Westerlind et al. [3] found those who experience more frequent pain had a significant increase (*p* < 0.001) of depression at 5 years post-stroke. Further to this, depression was a significant explanatory factor for experiencing pain (Odds Ratio = 8.0). The significant finding by Westerlind et al., [3] was rated to have moderately low risk of bias whilst the significant result from Choi-Kwon et al. [25] was rated to have a high risk of bias and the significant findings from Kong et al. [28] and Naess et al. [31] were rated as moderately high. 

##### Depression and QOL, Qualitative Evidence

The effect of depression on QOL was not directly discussed in qualitative studies.

##### Evidence Summary

Findings suggest that depression negatively impacts QOL and pain can be a predictor of depression 5 years post-stroke. However, due to the mixed quality of studies, this conclusion should be viewed with caution. Further high-quality studies are needed to confirm findings.

#### 3.4.3. Anxiety and Pain

##### Scales used to assess anxiety

Two studies considered the association between anxiety and pain. Anxiety was represented by the Short Anxiety Inventory Scale (SAIS, *N* = 1), the Stroke Specific Anxiety questionnaire (*N* = 1) and the anxiety subscale of the HADS (*N* = 1).

##### Anxiety and Pain, Quantitative Evidence

Galligan et al., [26] found that pain was significantly and positively associated with health anxiety (no statistical or numerical change provided). Tang et al. [33] also found that pain was significantly associated with anxiety (*p* = 0.003). Further to this, Widar and Ahlstrom [35], using the Multi-Dimensional Pain Inventory Score (MPI-S), identified a significant positive association (*r* = 0.52) between pain severity and affective distress (*p* < 0.01). The significant finding by Tang et al. [33] was rated to have moderately low risk of bias. The significant findings by Galligan et al. [26] and Widar and Ahlstrom [36] were rated to have moderately high risk of bias. 

##### Anxiety and Pain, Qualitative Evidence 

Qualitative studies found that patients feared side effects of medication and polypharmacy [29,36]. Individuals with tension-type headaches feared that a new clot was forming [36]. Pain was also reported to be worrying [35] and said to heighten emotions [29], which may be associated with anxiety being a higher energy negative emotion. Further to this, anxiety was also identified as a side effect of pain medication [29]. 

##### Evidence Summary 

The level of bias identified in two quantitative studies means that the association between anxiety and pain should be interpreted with caution. The qualitative data adds weight to the quantitative findings by identifying fear and worry as emotions that influence pain. Taken together, there is some evidence that anxiety is likely to have an adverse effect on the experience of pain; however, more high-quality studies are needed to understand the association further.

#### 3.4.4. The Effect of Anxiety on QOL

The effect of anxiety on QOL was not directly discussed in studies included for this review. 

#### 3.4.5. Fatigue

##### Scales Used to Assess Fatigue

Four studies considered the association between fatigue and pain using the Fatigue Assessment Scale (FAS, *N* = 1), single-item Likert Scale (*N* = 1) and Fatigue Severity Scale (FSS, *N* = 3). 

##### Fatigue and Pain, Quantitative Evidence

Naess et al. [30] found a significant and positive association (*r* = 0.27) between pain and fatigue (*p* < 0.001), as did Tang et al. [33] (*p* < 0.001), both on the FSS. Galligan et al. [26] also found this association (*r* = 0.35, *p* < 0.001) on the FAS. Further to this, Naess et al. [31] found a significant and negative association (*r* = −0.44) between pain (as measured by EuroQol-Visual Analogue Scale; EQ-VAS) and fatigue (*p* < 0.001) (on the FSS). Furthermore, there was a significant positive association (*r* = 0.21) between fatigue and sleep disturbances (*p* < 0.05) [26]. The significant result from Naess et al. [30] and Tang et al. [33] were of moderately low risk of bias; however, the significant results from Galligan et al. [26] and Naess et al. [31] were of moderately high risk of bias.

Further studies commented on the association between fatigue and pain, although specific fatigue outcome measures were not used. Severity of pain increased at night according to a Visual Analogue Scale (VAS) (no statistics provided) [32]. Furthermore, pain was found to disturb sleep in over half of the participants, as measured by a self-reported assessment of pain characteristics [27]. In contrast, pain was found to have no interference with sleep (*p* = 0.69) according to a pain questionnaire [1]. 

##### Fatigue and Pain, Qualitative Evidence

Among the qualitative studies, patients reported the experience of pain as exhausting [36] and that pain led to fatigue [29]. Fatigue was said to be worse during the night [29]. Increased tiredness was reported to be a side effect of pain medication [29], while medication was also taken to aid in sleeping [37]. 

##### Evidence Summary

Given the above findings, fatigue is thought to increase the experience of pain; however, considering the mixed quality among studies, further high-quality studies are needed to fully understand the association. The qualitative data adds weight to the quantitative findings by identifying the experience of pain as well as the side effects of medication as fatigue-inducing. The physical and emotional energy used in dealing with pain may lead to fatigue. Pain is likely to also have an adverse behavioural outcome on sleep which explains the need for medication to aid sleeping. Disrupted sleep would likely negatively affect energy levels, further adding to fatigue.

#### 3.4.6. The Effect of Fatigue on QOL

##### Scales Used to Assess QOL

One study considered the association between fatigue and HRQOL using the EQ-5D (*N* = 1) and 15D (*N* = 1).

##### Fatigue and QOL; Quantitative Evidence

Naess et al. [31] found no association between fatigue and QOL, as measured by EQ-5D; however, a significant negative association (*r* = −0.34) was found between fatigue and QOL when measured by the 15D (*p* < 0.001). Further to this, sleep disturbances were found to have a significant negative association (*r* = −0.37) with QOL, as measured by the 15D (*p* < 0.001). These contrasting findings indicated a need for a more specific measure of fatigue on QOL.

##### Fatigue and QOL, Qualitative Evidence

The effect of fatigue on QOL was not directly discussed in qualitative studies

##### Evidence Summary

Naess et al. [31] was the only study included in this review to comment directly on the relationship of these domains. Where an association was found between fatigue and QOL using the 15D, a sleep domain was included in the outcome measure. Where no association was found using the EQ-5D, no fatigue or sleep domain was included, making the outcome measure inadequate. The study had moderately high risk of bias and therefore findings should be interpreted with a high-level caution. The current understanding is limited, although suggests that fatigue negatively affects QOL. Further research is required to fully establish the impact.

#### 3.4.7. Cognitive Function and Pain

##### Scales Used to Assess Cognitive Function

Three studies considered the association between cognition and pain using the Mini-Mental State Examination (MMSE, *N* = 2) and the Montreal Cognitive Assessment (MoCA, *N* = 1).

##### Cognitive Function and Pain, Quantitative Evidence

Jonsson et al. [27] found a significant positive association between pain intensity and less cognitive decline (a higher MMSE score) (*p* = 0.004). Conversely, Tang et al. [33] found a significant association between pain intensity and increased cognitive decline (a lower MMSE score) (*p* = 0.038). Galligan et al., [26] found that pain caused catastrophising, hypervigilance and biased attention to somatic symptoms (no statistical or numerical change provided). Widar and Ahlstrom [35], however, found no association between pain severity and distracting responses. The studies by Jonsson et al. [27] and Tang et al. [33] were rated as moderately low risk of bias, whilst the studies by Galligan at al. [26] and Widar and Ahlstrom [36] were rated as moderately high risk of bias.

##### Cognitive Function and Pain; Qualitative Evidence

Biased attention to somatic symptoms was supported by qualitative studies in that patients distracted themselves from thinking about the pain by doing activities [29,36]. Catastrophising was also supported by a qualitative study in patients that experienced headaches, “a few imagined the possibility that the infarct was still present and creating pressure in the head” [36]. Incomprehensibility was also commonly reported, where there was a lack of understanding about the cause of the pain [29,36].

##### Evidence Summary

Cognition is a challenging domain to assess because cognitive impairment affects the reliability of self-reported pain. Given the mixed quality of evidence, these findings should be interpreted with caution. Stroke is a traumatic life experience with cognitive changes that make patients fear deterioration in their health and future.

#### 3.4.8. The Effect of Reduced Cognitive Function on QOL

##### Scales Used to Assess QOL

One study considered the association between cognitive function and QOL using the mental component summary (MCS) of the Short form-12 (SF-12).

##### Cognitive Function and QOL, Quantitative Evidence

Tang et al. [33] found a significant negative association (*r* = −0.15) between pain and MCS (*p* < 0.071). 

##### Cognitive Function and QOL, Qualitative Evidence

In a qualitative study, retaining cognitive capacity and being able to communicate with others was identified as an important aspect of QOL [35].

##### Evidence Summary

Findings suggest that reduced cognitive function negatively impacts QOL, however, given the limited evidence, further research is needed to be able to conclude this.

#### 3.4.9. Physical Function and Pain

##### Scales Used to Assess Physical Function

Five studies considered the association between physical function and pain using the Barthel Index (BI, *N* = 5), Modified Barthel Index (MBI, *N* = 1), the activities of daily living (ADL) staircase (*N* = 1), the Instrumental ADL (IADL) (*N* = 1), the Modified Rankin Scale (MRS, *N* = 2), The National Institute of Health Stroke Scale (NIHSS, *N* = 2), the Motor Impairment: Medical Research Council Motor Scale (*N* = 1), the Functional Independence Measure (FIM, *N* = 1) and the Risk of Stroke Questionnaire (*N* = 1).

#### Physical Function and Pain; Quantitative Evidence

Jonsson et al. [27] found that pain negatively affects physical function, where a significant association was found between a higher NIHSS score (decreased function) and pain intensity, as measure by the VAS (*p* < 0.001). The study also found females to have more impaired functional status (lower BI) (*p* = 0.038) and a higher pain intensity (*p* = 0.006) than males. Widar and Ahlstrom [36] supported these findings by identifying a significant positive association (*r* = 0.68) between pain severity and the level of interference of pain on function, according to the MPI-S scale (*p* < 0.01). Westerlind et al. [3] reported that more restricted movement was positively associated with the experience of pain when compared between those who experienced more and less pain (*p* = 0.002). Indeed, having restricted movement was an explanatory factor for experiencing pain (odds ratio = 3.6). Further, those people who were functionally dependent at discharge reported more frequent pain ( *p*= 0.018) and had higher odds of experiencing more frequent pain at 5 years (odds ratio = 2.4).

Tang et al. [33] found no association between pain and physical function. Kong et al. [28] also found no association between pain and physical function; however, an MCID was identified. The MCID for the BI is 1.85 points as determined by Hsieh et al. [40]. Results showed an MCID difference of 2.7 points between patients with and without pain (based on the mean score).

Sahin-Onat et al. [32] found mixed results. No association was found between pain and physical function, as measured by the FIM. Severity of pain decreased with activity. However, to contradict this finding, a significant negative association was found between pain and physical role limitation (*p* < 0.05) and physical score (*p* < 0.01) domains of the SF-36. Furthermore, there was a significant negative association (*r* = −0.619) between the Leeds Assessment of Neuropathic Symptoms and Signs scale (LANSS) and physical score of the SF-36 (*p* = 0.0001).

Jonsson et al. [27], Tang et al. [33], Sahin-Onat et al. [32] and Westerlind et al. [3] were rated as having a moderately low risk of bias. Kong et al. [28] was rated as having moderately high risk of bias.

#### Physical Function and Pain, Qualitative Evidence

Qualitative studies supported the findings from Jonsson et al. [27] and Sahin-Onat et al. [32], where pain was found to affect physical function by limiting movement. Pain was induced by certain positions and meant activity limitations as well as adaptation of activities to complete ADLs [29,36]. Widar et al. [37] also found that physical exertion provokes headaches whilst Lindgren et al. [29] found that some participants experienced more pain when at rest.

#### Evidence Summary

Findings from the higher quality studies suggest mixed findings. It is therefore difficult to conclude. It is evident that pain affects physical function, but is unclear whether physical function negatively or positively affects the experience of pain. Findings by Sahin-Onat et al. [32] suggest that pain affects physical function and negatively influences physical activity; however, these conclusions were drawn from the SF-36, a measure of QOL, rather than a specific measure of function. It, therefore, also identifies the need for a more specific measure of pain and function to truly understand the link. The qualitative data add weight to this conclusion in that limited movements can compromise gait and balance, which impact the ability to complete ADLs. The finding by Widar et al. [37], that physical exertion provokes headaches, is arguably contradicted by Lindgren et al. [29], where participants experienced increased pain at rest.

#### 3.4.10. The Effect of Reduced Physical Function on QOL

##### Scales Used to Assess QOL

Three studies considered the association between physical function and QOL using the Barthel Index (BI, *N* = 2) and the SF-36 (*N* = 1) or the SF-12 (*N* = 1) and EQ-5D (*N* = 2).

##### Physical Function and QOL, Quantitative Evidence

Naess et al. [31], found a significant positive association (*r* = 0.55) between the BI and QOL, as measured by the EQ-5D (*p* < 0.001). Westerlind et al. [3] found significantly worse outcomes across usual activities, self-care and mobility for people who reported more pain compared with those who reported less pain (*p* < 0.001). Choi-Kwon et al. [25] found that being dependent in ADLs was significantly and negatively associated with QOL (*p* < 0.01). A small effect size was calculated amongst patients with a BI<96 (Hedges’*g* = (3.16–3.05)/0.49 = 0.23). A very small to small effect size was calculated amongst patients with a BI ≥ 96 (Hedges’*g* = (3.51–3.58)/0.44 = 0.16) [39]. A significant negative association was also found between motor dysfunction and QOL (*p* < 0.01). Tang et al. [33] found a significant negative association (*r* = −0.260) between pain and Pain Catastrophizing Scale (PCS) (*p* < 0.001). Tang et al. [33] was rated as moderately low risk of bias. Naess et al. [31] was rated as moderately high risk of bias, whilst Choi-Kwon et al. [25] was rated as high risk.

##### Physical Function and QOL, Qualitative Evidence

A qualitative study reported that retaining physical capacity and independence was determined as an important aspect of QOL [35]. 

##### Evidence Summary

Given the above findings, the qualitative evidence adds weight to the significant quantitative findings, suggesting that reduced physical function negatively impacts QOL. However, the low-quality evidence indicates a need for more research of higher quality studies to confirm these findings.

#### 3.4.11. The Effect of Pain on QOL

##### Scales Used to Assess QOL

Five studies considered the association between pain and QOL using the SF-36 (*N* = 3), EQ5D (*N* = 1), 5D (*N* = 1), WHOQOL (*N* = 1) and multi-dimensional pain inventory scale (MPI-S, *N* = 1)

##### Pain and QOL, Quantitative Evidence

Pain was found to negatively affect QOL in three studies. Naess et al. [31], found a significant negative association between pain and QOL, both on the EQ-5D (*r* = −0.57, *p* < 0.001) and the 15D (*r* = −0.53, *p* < 0.001). Widar et al. [35] found participants had a deteriorated HRQOL, as a consequence of pain. Different types of pain (CPSP vs nociceptive vs headache) had no significance. Choi-Kwon et al. [25] found the presence of CPSP to be significantly and negatively associated with QOL at three months (*p* < 0.01) and three years (*p* < 0.05) post-stroke, as measured by the WHOQOL. The effect size was zero where sensory symptoms were absent (Cohen’s *d* = (3.48–3.48)/0.46 = 0). A very small effect size was calculated where moderate parathesia was present (Hedges’*g* = (3.36–3.38)/0.36 = 0.06) and a very small effect size was calculated where CPSP was present (Hedges’*g* = (2.93–2.91)/0.46 = 0.04) [39]. This finding showed that neuropathic pain negatively affects QOL, regardless of its severity.

In contrast, two studies found that pain has no effect on QOL [28,32]. The significant findings by Naess et al. [31], Widar et al. [35], and Widar et al. [34] were rated as moderately high risk of bias. Choi-Kwon et al. [25] was of high risk.

##### Pain and QOL, Qualitative Evidence

Qualitative studies identified freedom from pain as an important aspect of QOL [35].

##### Evidence Summary 

The findings discussed above suggest that pain has a negative effect on QOL. This is further supported by the qualitative evidence. However, the low-quality of findings means that evidence cannot permit a conclusion. Further research of high-quality studies is needed to understand the impact of pain on QOL.

##### Interactions and Associations 

Many of the components analysed in these studies revealed associations across domains. Galligan et al. [26] found a significant positive association between fatigue and health-related anxiety (*r* = 0.31) and stroke-specific anxiety (*r* = 0.37). This could be related to the findings discussed earlier regarding catastrophising [26], heightened emotions [29] and fear [36] that pain causes. Further to this, a significant positive association was found between fatigue and depression by both Galligan et al. [26] (*r* = 0.225, *p* < 0.05), and Naess et al. [31] (*r* = 0.49, *p* < 0.001). These high energy emotions can lead to fatigue, as identified by Widar et al. [34].

Galligan et al. [26] found a significant positive association between fatigue and functional impairment (*r* = 0.244, *p* < 0.05). This finding was supported in a qualitative study which showed previously that active participants adapt to physical activities because they are too physically demanding and they need to accommodate their changed physical capabilities [36].

A relationship was also found between functional impairment and psychological distress (*r* = 0.26, *p* < 0.01), depression (*r* = 0.26, p < 0.05), general anxiety (*r* = 0.24, *p* < 0.05) and stroke specific anxiety (*r* = 0.24, *p* < 0.05) [26].

Co-occurring symptoms were found to heighten pain. Naess et al. [30] found pain (as determined by VAS), fatigue (as determined by FSS) and depression (as determined by HADS) severity increased with the number of co-occurring symptoms (pain, fatigue and depression). There was a significant negative association (*r* = −0.33) between the number of symptoms and the BI (*p* < 0.001); the more symptoms, the more difficulty with ADL’s. This further supports the associations between domains that have been found.

## 4. Discussion

To the best of the authors knowledge, this was the first review to integrate findings around PSP and QOL. The review identified the primary factors that affect the experience of PSP and considers the impact of these factors on QOL. Further discussion is provided according to the main outcomes.

Chronic pain following stroke is prevalent in 32–45% of patients between three months and two years post-stroke [1,27,33]. Findings highlight that the experience of pain is multifactorial and there are associations between the identified factors. 

### 4.1. Emotional Distress

#### 4.1.1. Depression

Depression was consistently identified as having a negative effect on the experience of PSP by increasing its intensity [1,27,33]. Depression increases the perception of pain [29] and causes patients to have increased feelings of anger when experiencing pain [36]. Depression can be a cause of pain as well as occurring as a response to pain. This agrees with the biopsychosocial model by Gatchel et al. [41], who states that the link between pain and emotional distress is bidirectional.

The significant finding by Jonsson et al. [27] used the GDS, intended for a geriatric population (over 65) [42]. The mean age of participants in this sample had a range of 17–96 years. This outcome measure was therefore not representative of the whole sample used, as it excluded questions related to the symptoms of depression that are likely to be common in people under the age of 65, and an inaccurate reflection of these participants.

Despite these notable findings, other studies failed to find an association between depression and PSP [26,28,30,32]. However, one of these studies [32] had an MCID between pain and depression (as measured by the BDI); depression affects the experience of pain enough to have a clinical effect on the patient despite there being no statistical significance. The lack of association found by Kong et al. [28] could be explained by the fact that 57% of patients had sensory impairments and 11% had neglect, both of which can alter perceived pain [1]. Further to this, pain was measured by the BPI which has not been validated as an outcome measure within the stroke population [28]. Galligan et al. [26] reported no association between depression and pain as individual variables, but both depression and pain adversely affected fatigue and anxiety. Naess et al. [30] found similar findings, highlighting that there is overlap between factors. Considering these explanations and the quality of evidence, depression likely has a negative effect on the experience of PSP. This is supported by other research [43] that identified a positive association between PSP and agitation, aggression and depression amongst a population of 274 stroke patients, where the prevalence of PSP was 58%.

#### 4.1.2. Anxiety

Anxiety was found to negatively affect the experience of PSP in both studies that assessed this association [26,33]. This again agrees with past findings [43], in which PSP was positively associated with anxiety. To support this, Widar and Ahlstrom [36] also identified that pain severity increases affective distress, however this finding was identified using the MPI-S, a generalised pain outcome measure not specific to anxiety.

Fear is defined as anxiety attached to a circumstance [44], and is present in stroke patients in that they fear side effects of medication [29,37] and those with tension-type headaches fear the formation of new clots [37]. Pain was also described as worrying [36]. Some patients may have a constant need for reassurance, while others may avoid seeking medical attention through fear of discovering that something is wrong. Furthermore, pain is also said to heighten emotions [29]; anxiety is a higher energy negative emotion that is also associated with post-stroke fatigue (PSF) as highlighted by Galligan et al. [26]. Anxiety also impacts cognitive functioning by reducing focus on a task [45]. These highlight the overlap between factors.

### 4.2. Other Factors

#### 4.2.1. Fatigue

Fatigue is associated with emotional distress. This connection is acknowledged by the national stroke guidelines, stating patients with PSF should be screened for depression [46]. Fatigue was found to have a negative effect on the experience of PSP [26,30,31,36] and determined as a symptom of emotional distress and pain [26,31]. Fatigue was identified as the most important clinical variable to have an influence on the experience of PSP [31]. These findings were supported by Hinkle et al. [47] and Miller et al. [48], that looked specifically at PSP and fatigue.

Brain injury requires new motor learning through neuroplasticity [49]. New adaptations to movement, due to pain and physical deficits, use more energy due to using different, less efficient muscles. These factors contribute to fatigue, particularly in already deconditioned older patients. This explains findings by Galligan et al. [26], where fatigue was found to increase functional impairment.

Conflicting results were found with regard to whether pain interferes sleep. Klit et al. [1] reported no association according to a pain questionnaire. However, Galligan et al. [26] using the FSS, a specific measure of fatigue, found that pain causes sleep disturbances, possibly a result of emotional distress, cognitions and physical pain during the night. Sleep disturbances cause hyperalgesia [50], which further supports the adverse effect that fatigue has on the experience of pain. Recovery sleep was found to restore pain sensitivity by Stroemal-Scheder et al. [50]. However, if sleep is continuously disturbed, pain sensitivity remains heightened.

Severity of pain was reported to increase at night [29,32] possibly due to biased attention to symptoms as cognitive distractions are reduced [26]. The negative behavioural outcome that pain has on sleep further adds to fatigue by negatively affecting energy levels. 

#### 4.2.2. Cognitive Function

Galligan et al. [26] found that pain causes catastrophising, hypervigilance and biased attention to somatic symptoms. This finding was supported by the cognitive behavioural model [51], stating that the experience of pain and illness activates physiological mechanisms that further increase pain, attention and avoidance.

Decreased cognitive function was associated with an increased experience of pain [33]. To oppose this, Jonsson et al. [26] found that an increased cognitive function was associated with an increased experience of pain, possibly because patients with less of a cognitive decline are more physically active and therefore do more pain provoking movements compared with cognitively impaired patients. Various mechanisms may explain these findings on cognitive function. For instance, cognitive function could be reduced or influenced by; (a) constant nociceptive inputs caused by CPSP, as they compete with other sensory inputs [52] or (b) neuroplastic changes in the brain as a response to CPSP [53].

Both studies evaluated cognition using the MMSE, a test of global cognition with low sensitivity and specificity [54]. Across qualitative studies, incomprehensibility was commonly reported with patients not understanding the cause of their pain [29,37], possibly feeding into the fear, anxiety, attention and catastrophising [26,37]. Cognition is a challenging domain to assess because cognitive impairments affect the reliability of self-reported pain; therefore, given the mixed findings in this review, the effect of cognitive function on the experience of PSP remains unclear.

#### 4.2.3. Physical Function

Physical function was found to be negatively affected by pain [26,34] in that the presence of pain causes reduced physical function. Pain can cause limitations to activities due to avoidance behaviours and result in patients finding adaptive ways of completing ADL’s [29,36]. The cognitive behavioural model states that adaptation of activities and avoidance behaviours can lead to further sensitisation through operant conditioning [51], suggesting an increase in pain experience.

Sahin-Onat et al. [32] found conflicting results. No association was found between pain and physical function, as measured by the FIM; however, pain was found to increase two physical aspects of the SF-36; physical score and physical role limitations score. The SF-36 although not a specific measure of function, is arguably a better evaluation where tasks are more physically demanding (carrying shopping, lifting heavy objects and, running) compared with the FIM that measures basic ADLs (feeding, dressing and transfers).

These findings were not consistent in other studies where no association was identified [28,33], however findings by Kong et al. [28] had an MCID for the BI suggesting a meaningful change in physical function from the patient’s perspective. Furthermore, the lack of association identified by Tang et al. [33] could be possibly because patients that have higher physical function are more physically active and therefore do more pain provoking activities.

Widar et al. [36] found that physical exertion provokes headaches. Headaches can be a consequence of physical exertion due to constriction of blood vessels around the neck and head, and increase in blood pressure [55]. Lindgren et al. [29] found that participants experienced more pain when at rest, when there is heightened awareness and attention to pain because of a reduced cognitive distraction, as previously discussed [26,51].

Physical deficit following stroke can lead to anxiety and fear of falling which cause avoidance of activities and reduced tolerance to activity, resulting in further decline. The lack of physical activity and consequently increased dependency on others can be associated with low mood due to loss of independence [29,35], relationship strain [35,36] and social isolation [29,36].

Findings therefore suggest that PSP causes reduced physical function, agreeing with past research [48], which identified that pain correlated with physical activity and adversely affected rehabilitation outcomes. However, due to the nature of this domain, it remains unclear whether physical function negatively or positively affects the experience of pain.

### 4.3. Impact on QOL

#### 4.3.1. Depression

Depression was found to negatively impact QOL [25,28,31]. Depression was found to be a predictor of low QOL by Choi-Kwon et al. [25]. However, the incidence of depression was only 13% at three months post-stroke and 5% at three years post-stroke. This low incidence that further reduced could be because almost all participants lived at home with spouses or children, which Anderson et al. [56] identified as a favourable environment for depressed patients.

Naess et al. [31] found that depression negatively affects HRQOL when measured by the 15D, but not when using the EQ-5D; however, the 15D assesses both depression and distress as separate domains, while the EQ-5D assesses anxiety and depression as a single domain, making 15D a more appropriate outcome measure in this instance. These findings correlate with past research [57], which found that depression was associated with lower QOL.

#### 4.3.2. Fatigue

Fatigue was found to negatively impact QOL [31]; however, only one study in this review assessed the association. The study found that fatigue, as measured by the FSS, negatively affected QOL when measured by the 15D, but had no association when measured by the EQ-5D. The 15D also identified that sleep disturbances negatively impact QOL. This disparity between results could be because the 15D includes questions about fatigue, while the EQ-5D does not. The EQ-5D is therefore, an inadequate outcome measure to use in determining whether fatigue impacts QOL. The adverse effect that fatigue has on QOL is supported by past research [48], which found that PSF leads to decreased participation in physical activities and rehabilitation, and difficulty returning to social and professional activities, resulting in lower QOL scores. 

#### 4.3.3. Cognitive Function

Retaining cognitive capacity and being able to communicate with others was identified as important aspects of QOL [35]. This could be related to the social isolation and lack of social support that patients experience when cognitive impairment impairs their ability to communicate and participate in social and physical activities. Tang et al. [35] found that those with reduced cognitive capacity (as measured by the MCS element of the SF-36) experienced more pain and a reduced QOL. However, as previously discussed, cognition is a challenging domain to assess and further studies are needed to understand how altered cognitive function impacts QOL.

#### 4.3.4. Physical Function

Retaining physical capacity and independence was identified as an important aspect of QOL [36]. This is in line with Dysvik et al. [10], who identified that each of the subscales of the SF-36 were lower where pain was present; the greatest effect was on physical role, referring to limitations in daily activities. Lack of independence can cause depression which can lower motivation to be physically active [58]. The ability to participate in physical activity can be affected by associated fatigue, depression and psychological distress [26]. In a systematic review by Billinger et al. [59], exercise was found to positively impact QOL of stroke patients. PSF was found to be aggravated by a sedentary lifestyle. 

#### 4.3.5. Pain 

The overall experience of pain appeared to have a consistent and negative impact on QOL. Several studies support this finding [8,9,10]. Pain was reported to have no impact on QOL [28] when the BPI was used, an outcome measure that has not been validated for use in a stroke population. Furthermore, the mean MBI scores of the sample used were high (96.5/100), indicating that a high proportion of the sample were functionally independent and may therefore experience less pain. Therefore, this makes the findings unrepresentative of a stroke population and lowers validity of findings.

Symptom clusters (two or more symptoms) are common [30,31] and co-occurring symptoms further influence the experience of pain by increasing its intensity [30]. There is also overlap between symptoms, highlighting the need for individual symptoms to be recognised and treated to reduce the negative impact on QOL.

### 4.4. Implications 

Pain has emotional and behavioural consequences that impact QOL. The identified factors are not routinely assessed post-stroke. An objective measure of PSP that also considers the five primary factors could be used to recognise the presence of these symptoms among patients and inform a treatment plan to reduce the negative impact of these factors on both PSP and QOL.

It is important to educate clinicians to have increased awareness and understanding of PSP and factors that influence it to ensure better recognition and subsequent treatment.

It is important to educate patients about reasons they may experience pain and provide reassurance to reduce negative feelings, anxiety and depression, that increase the experience of pain. Further education about the benefits of physical activity and negative effects of immobility may also have positive outcomes. 

### 4.5. Limitations

A limitation of this review was that it included studies that looked at PSP and QOL up to three years post-stroke which is not longitudinal enough to represent the increased effect that stroke has on QOL as patients age, indicating that longer term follow-up studies are needed. Furthermore, the studies looked at PSP at different time points post-stroke; the influence of these factors may differ at different time points.

Seven studies excluded patients with cognitive impairments and communication issues, common in a stroke population and therefore not representative [25,26,28,33,35,36]. Where patients with cognitive impairment were not excluded, the reliability of self-reported questionnaires was compromised [27].

The quality of studies was assessed by the primary researcher. To improve the reliability of findings, quality assessment should be carried out by a secondary researcher, as appraisal relies on subjective judgement.

There is a risk of response bias in all included studies due to factors being self-reported and highly subjective, although self-report is the gold standard for assessment of pain [60].

Despite some studies being low quality, the risk of bias tool assesses studies as if they were in a controlled environment which is impractical within this field of research; therefore, conclusions made from low-quality studies should still be considered and were included in the discussion.

Several outcome measures were used across studies. For increased validity, further research should use the same outcome measure across domains.

Comorbidities make it challenging to differentiate the relative influence of the identified factors on PSP and QOL.

This review explored the effects of five primary factors. Further research is needed to consider the possible effects of social, environmental and religious factors on the experience of PSP and QOL. 

## 5. Conclusions

The experience of pain is not a reflection of nociceptive input and is multifactorial in that it is influenced by many factors. In order to optimize QOL and function post-stroke, clinicians need to be aware of PSP, the factors that influence it and the difficulty that patients may have in communicating it. Early identification could improve patients’ QOL.

The results of the studies reviewed provide evidence that anxiety, depression and fatigue heighten pain. Further research is required to assess how cognitive function and physical function affect the experience of pain. It is also evident that pain, depression, fatigue and reduced physical function have a negative impact on QOL. Further research is needed to establish how anxiety and reduced cognitive function impact QOL and at what stage post-stroke these factors influence PSP. This can aid the development of an outcome measure to identify and objectively quantify PSP and its impact on QOL.

## Figures and Tables

**Figure 1 behavsci-10-00128-f001:**
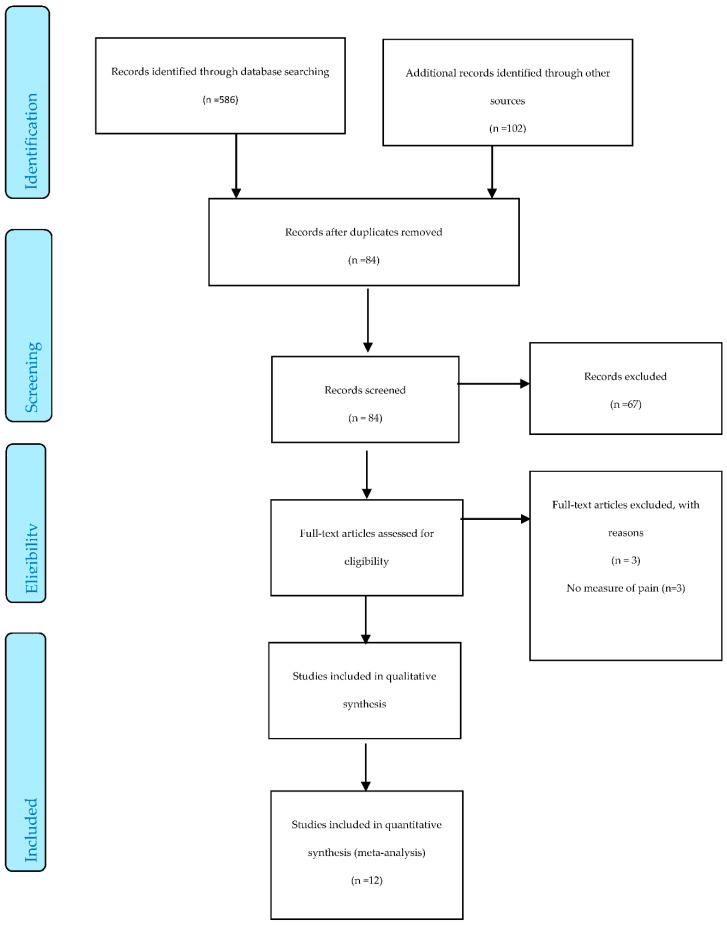
The PRISMA Flow Diagram.

**Table 1 behavsci-10-00128-t001:** The association between post stroke pain and key factors; an overview of evidence.

Factor	Quantitative Evidence	Qualitative Evidence	Summary
Depression and PSP	4 studies (quality rated as low risk) found no association, but in 1 study a clinically important difference was identified. 4 studies (quality rated as low to moderate risk) identified significant positive association between depression and PSP.	Pain negatively influences mood and generate disheartening experiences. Pain could make people angry.	Mixed findings exist. Very possible that PSP negatively impacts low mood. Possible that PSP is associated with frustration and anger.
Depression and QOL	4 studies (quality rated as high risk) identified a significant negative association between depression and QOL.		Depression likely has a negative impact on QOL. Possible that pain can be a predictor of depression 5 years post-stroke.
Anxiety and PSP	3 studies (quality rated as moderate risk) identified a significant positive association between anxiety and PSP.	Pain heightens emotions Pain could make people fearful of their future health, further complications or suffering another stroke	Anxiety likely has a negative effect on pain. Fear and worry as emotions are also likely to have a negative association with PSP.
Fatigue and PSP	4 studies (quality rated as moderate risk) identified a significant positive association between fatigue and PSP.	Pain is an exhausting experience and leads to fatigue Fatigue can be a side effect of medication	Fatigue is likely induced by the experience of pain. The emotional energy used to deal with pain also likely contributes to fatigue Pain likely negatively influences sleep quality further contributing to fatigue. Fatigue is a possible result of physical decline and emotional distress
Fatigue and QOL	1 study (quality rated as moderate risk) identified a significant negative association between fatigue and QOL.		Limited evidence but possible that fatigue negatively affects QOL.
Cognitive function and PSP	1 study (quality rated as low risk) identified a significant positive association between cognitive function and PSP. 1 study (quality rated as low) identified a significant negative association between cognitive function and PSP.	PSP can cause biased attention to somatic symptoms. PSP can result in people catastrophising Incomprehensibility as a result of not understanding the cause of pain	Mixed findings exist Cognitions influence how pain is experienced in that interpretations catastrophising and attention affects beliefs about PSP.
Cognitive function and QOL	1 study (quality rated as low risk) identified a significant negative association between cognitive function and QOL	Retaining cognitive capacity was identified as an important aspect of QOL.	Limited evidence but possible that cognitive function negatively impacts QOL.
Physical function and PSP	3 studies (quality rated as low risk) identified a significant negative association between physical function and PSP. 2 studies (quality rated as moderate risk) found no association between physical function and PSP, but in 1 study a clinically important difference was identified.	PSP limits movement. PSP can be the cause of adaptation of activities.	Mixed findings exist. Challenging to distinguish between the cause and effect between pain and physical. function. Reduced physical function could be a result of pain, whilst increased physical function may provoke pain.
Physical function and QOL	3 studies (quality rated as high risk) identified a significant negative association between physical function and QOL.	Retaining physical function and independence was identified as an important aspect of QOL.	Reduced physical function likely negatively impacts QOL.
Interactions and associations	1 study (quality rated as moderately high risk) identified a significant positive association between anxiety and fatigue. 2 Studies (quality rates as moderately high risk) identified a significant positive association between fatigue and depression. 1 study (quality rates as moderately high risk) identified a significant positive association between fatigue and functional impairment.		Associations exist across domains. Co-occurring symptoms can further heighten the experience of PSP.

Note: QOL = Quality of Life, PSP = Post Stroke Pain.

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
