# Peer review of "The Experience of Post-Stroke Pain and The Impact on Quality of Life: An Integrative Review"

_behavsci, 2020, doi:10.3390/bs10080128_

Round 1
Reviewer 1 Report
This review entitled "The experience of post-stroke pain and the impact on quality of life: an integrative review", Payton and Soundy aimed at reviewing current knowledge on PSP and its impact on quality of life (QOL). Authors used a 4 steps process to select studies to include in their review (Identification-Screening-Eligibility-Inclusion). I found their methods and criteria appropriate. I also found the review balanced and very informative. Some minor points that authors should address are the following:
- I would speculate on the potentially involved mechanisms that may deserve research attention to link cognitive function and pain and physical function and pain.
- Figure 1 is not of publication quality. Please hide formatting marks.
- I suggest to revise the manuscript for typos
Author Response
- I would speculate on the potentially involved mechanisms that may deserve research attention to link cognitive function and pain and physical function and pain.
Reply: We have identified some mechanisms within the discussion.
- Figure 1 is not of publication quality. Please hide formatting marks.
Reply: We have updated the figure and adjusted box sizes.
- I suggest to revise the manuscript for typos
Reply: We have revised and considered typos.
Reviewer 2 Report
Overall a good meta-analysis on topic which important, however has not been explored extensively.
I would suggest to revisit Figure 1. The PRISMA Flow Diagram, and make sure than numbers are accurate. For eg: Eligibility 4 states "study included n=2" and next section in table states "study included n=12". Which does not make sense. Appears to be typing error.
Author Response
Overall a good meta-analysis on topic which important, however has not been explored extensively.
Reply: Thank you for this comment.
I would suggest to revisit Figure 1. The PRISMA Flow Diagram, and make sure than numbers are accurate. For eg: Eligibility 4 states "study included n=2" and next section in table states "study included n=12". Which does not make sense. Appears to be typing error.
Reply: Thank you. The 2 and 12 refer to the quantitative and qualitative synthesis. We have underlined these elements in the boxes to make sure this is clear for the reader.
Reviewer 3 Report
In the manuscript entitled “The Experience of Post-Stroke Pain and the Impact on Quality of Life: An Integrative Review”, the objectives of the Authors are to provide an integrative review about the current knowledge regarding the pain occurred post-stroke. In particular, to better understand the primary factors that affect post-stroke pain and determine the impact on the quality of life, 14 studies were identified and 2415 (968 females, 1447 males) people were included.
The review is well-written and of high interest for the scientific community. However, the figure provided (Figure 1) is of low quality and are evident the sign of formatting; please provide the final version of this figure.
Further, to summarize the results of the 14 studies and increase the readability, it would be interesting to add a table with the different factors involved.
Author Response
The review is well-written and of high interest for the scientific community. However, the figure provided (Figure 1) is of low quality and are evident the sign of formatting; please provide the final version of this figure.
Reply: we have updated the figure and adjusted box sizes.
Further, to summarize the results of the 14 studies and increase the readability, it would be interesting to add a table with the different factors involved.
Reply: We have added table which summarises the results with factors and summary of evidence.